# Neither Trimethylamine-N-Oxide nor Trimethyllysine Is Associated with Atherosclerosis: A Cross-Sectional Study in Older Japanese Adults

**DOI:** 10.3390/nu15030759

**Published:** 2023-02-02

**Authors:** Jubo Bhuiya, Yoshitomo Notsu, Hironori Kobayashi, Abu Zaffar Shibly, Abdullah Md. Sheikh, Ryota Okazaki, Kazuto Yamaguchi, Atsushi Nagai, Toru Nabika, Takafumi Abe, Masayuki Yamasaki, Minoru Isomura, Shozo Yano

**Affiliations:** 1Department of Laboratory Medicine, Faculty of Medicine, Shimane University, 89-1 Enya-cho, Izumo City 693-8501, Japan; 2Metabolizumo Project, Faculty of Medicine, Shimane University, 89-1 Enya-cho, Izumo City 693-8501, Japan; 3Department of Neurology, Faculty of Medicine, Shimane University, 89-1 Enya-cho, Izumo City 693-8501, Japan; 4Department of Cardiology, Faculty of Medicine, Shimane University, 89-1 Enya-cho, Izumo City 693-8501, Japan; 5Center for Community-Based Healthcare Research and Education (CoHRE), Shimane University, 89-1 Enya-cho, Izumo City 693-8501, Japan

**Keywords:** L-carnitine, trimethylamine-N-oxide (TMAO), trimethyllysine (TML), atherosclerosis, intima–media thickness (IMT)

## Abstract

Recent evidence suggests that trimethylamine-N-oxide (TMAO), a metabolite of L-carnitine and choline, is linked to atherosclerosis and cardiovascular diseases. As TMAO content is very high in fish, we raised the following question: why do Japanese people, who consume lots of fish, show a low risk of atherosclerosis? To address this question, we investigated the effects of TMAO and other L-carnitine-related metabolites on carotid intima–media thickness (IMT). Participants were recruited from a small island and a mountainous region. Plasma L-carnitine, γ-butyrobetaine (γBB), TMAO, trimethyllysine (TML), eicosapentaenoic acid (EPA), and docosahexaenoic acid (DHA) levels were measured using liquid or gas chromatography–mass spectrometry. Plasma L-carnitine concentration was higher in men than in women. TMAO and TML were significantly higher in the residents of the island than in the mountainous people. In multiple linear regression analyses in all participants, TML showed a significant inverse association with max-IMT and plaque score (PS), whereas TMAO did not show any associations. In women, L-carnitine was positively associated with max-IMT and PS. TMAO was correlated with both EPA and DHA levels, implying that fish is a major dietary source of TMAO in Japanese people. Our study found that plasma TMAO was not an apparent risk factor for atherosclerosis in elderly Japanese people, whereas a low level of TML might be a potential risk. L-carnitine may be a marker for atherosclerosis in women.

## 1. Introduction

Cardiovascular disease (CVD), which includes coronary artery disease, stroke, and peripheral arterial disease, is one of the leading causes of death in Japan [1]. In addition, CVD survivors often need both acute and post-acute care, which results in an increase in the social burden [2]. It is therefore necessary to prevent CVD in order to realize healthy longevity.

Atherosclerosis is a key contributor to CVD [3]. Thus, the suppression of atherosclerosis is essential to prevent CVD and its complications. In the last decade, a unique metabolite, trimethylamine-N-oxide (TMAO), attracted much interest because many studies suggested that TMAO promoted atherosclerosis, as well as CVD, in humans and mice [4,5,6,7]. TMAO has been further shown to increase platelet activity and potentiate thrombogenesis, which might be other crucial risk factors for cardiovascular events [8].

TMAO is mainly produced through a unique cooperation between the gut microbiota and the host, which consists of three metabolic steps [4]: (1) a dietary intake of L-carnitine or phosphatidylcholine, (2) the conversion of them to trimethylamine (TMA) by the gut microbiota, and (3) the oxidation of TMA to TMAO by the flavin-containing monooxygenase 3 (FMO3) isoform in the liver [9,10,11,12]. Red meat and eggs are the major sources of L-carnitine and choline, either in a free or in an esterified form [13,14,15]. Thus, the overconsumption of these foods might increase the risk of developing CVD through the production of TMAO [15]. L-carnitine, however, may increase CVD risk through other pathways, in addition to TMA/TMAO production, since (1) the level of plasma L-carnitine has been found to influence CVD risk among those with high TMAO levels, and (2) L-carnitine supplementation has been found to accelerate atherosclerosis in a murine model without affecting TMAO levels [16,17]. Further, as fish is rich in TMAO [18], the deteriorating effect of high TMA/TMAO on atherosclerosis and CVD has been challenged due to the fact that Japanese people, who consume a lot of fish, show a low risk of CVD [19]. In this context, it is worth noting that recent evidence has indicated that other metabolites related to L-carnitine, such as γ-butyrobetaine (γBB) and trimethyllysine (TML), are also risk factors for CVD [20]. TML, a carnitine precursor, has been found to be an independent predictor associated with CV mortality [20], CVD risk [21], and acute coronary syndrome [22]. A higher amount of γBB, which is rapidly synthesized from dietary L-carnitine [23], has been linked to CV mortality and carotid atherosclerosis [20]. 

In order to obtain further evidence on the roles of L-carnitine-related metabolites in atherosclerosis, we conducted a cross-sectional study to examine whether there is an association between carotid intima–media thickness (IMT) and L-carnitine-related metabolites in elderly populations in Japan. In this study, we hypothesized that TMAO might not be a risk factor for CVD in our population but rather a simple marker of fish consumption. Thus, we compared two populations, one from a mountainous area and one from a remote island, where fish consumption, i.e., the dietary intake of TMAO, was expected to be different. 

## 2. Materials and Methods

### 2.1. Subjects

This study was part of a cohort study (Shimane CoHRE Study) conducted by the Center for Community-based Health Research and Education, Shimane University. This study was undertaken in collaboration with various counties located in rural areas of Shimane Prefecture, Japan [24,25,26]. In this study, we selected two counties, Kakeya and Oki, because they are located in a mountainous region and on an isolated small island, respectively, where fish consumption was expected to be different. The inclusion criteria of the participants in the present study were all individuals aged 35 or older who underwent health check examinations conducted in Kakeya and Oki counties in 2015. Although we invited all residents according to the criteria, the participants in this study were mostly over 50 years of age due to the highly aging society in the area. The exclusion criterion was the presence of severe disorders, such as advanced cancer and heart failure. Based on these criteria, a total of 364 subjects (142 men and 222 women) were enrolled in this study. Histories of smoking, hypertension (HT), diabetes mellitus (DM), and dyslipidemia (DL) were obtained through an interview and medication report. Blood pressure or fasting blood glucose measured on site was not included in the criteria for the diagnosis.

### 2.2. Ethics

Written informed consent was obtained from each participant. The study protocol was conducted in accordance with the Declaration of Helsinki and approved by the local ethics committee of Shimane University (#20190522-1). 

### 2.3. Data Collection

Intima–media thickness (IMT) in the carotid artery was measured using high-resolution, real-time ultrasonography with a 7.5 MHz transducer (Vivid I or LOGIQ e, GE, Tokyo, Japan). Measurements of the arterial wall thickness were performed on four segments of the bilateral carotid arteries: at 1.5 cm distal to the bifurcation in the internal carotid artery (S1), at the bifurcation (S2), at 0–1.5 cm (S3), and 1.5–3.0 cm (S4) proximal to the bifurcation in the common carotid artery. The maximal value among the 8 measurements on S1–S4 of the bilateral carotid arteries served as max-IMT in the present study. Eight measurements at S1–S4 of the bilateral carotid arteries were summed up as plaque score (PS) if they were 1.1 cm or more. 

Venous blood was collected after overnight fasting. The plasma sample was separated within 30 min of the blood being drawn and kept frozen at −80 °C until the measurement of carnitine-related metabolites, eicosapentaenoic acid (EPA), and docosahexaenoic acid (DHA).

### 2.4. Measurements of L-Carnitine-Related Metabolites

L-carnitine, BB, TMAO, and TML were measured using liquid chromatography–tandem mass spectrometry (LC-MS/MS-8050, SHIMADZU Co., Kyoto, Japan) as previously described [16]. Briefly, an internal standard solution contained 0.1 μmol/L d9-carnitine, d9-γBB, d9-TMAO, and d9-TML prepared in methanol/acetonitrile (15:85) and 0.1% formic acid. After obtaining 10 μL of the plasma and the standard of each metabolite (Wako Pure Chemical Industries, Ltd. Osaka, Japan), the samples were pipetted into a 2 mL centrifuge tube, and 300 μL of the internal standard solution was added. All samples were targeted to contain 0.1 μg/mL d9-carnitine, d9-γBB, d9-TMAO, and d9-TML (Cambridge Isotope Laboratories, MA, USA). The mixture was vortexed for 2 min, followed by centrifugation (10,000× *g* for 10 min at 4 °C). The supernatant was taken for LC-MS/MS analyses using SHIMADZU 8050 triple quadrupole mass spectrometer interfaced SHIMADZU high-performance liquid chromatography system (LC-30AD), an autosampler (SIL-30AC), and a column oven (CTO-20AC). The elution was carried out using a BEH HILIC column (2.1 mm × 150 mm; 3.5 μm, Waters) with a BEH HILIC VanGuard pre-column (2.1 mm × 5 mm; 1.7 μm Waters). The separation was the following gradient mobile phase consisting of a mixture of 15 mmol/L ammonium formate (A) and acetonitrile as a solvent (B): 0 min 70% B; 0–3 min 17% B; and 3–5 min 70% B. The following settings were used: a flow rate of 0.4 mL/min, a sample injection volume of 2 μL, and a column oven temperature of 35 °C. Measurements of the metabolites were carried out based on the standard curve (Appendix A). The coefficient of variation (CV) of the measurement was less than 5% for all the metabolites. EPA and DHA were determined using gas chromatography MS (SRL Co., Ltd., Hachioji, Japan). 

### 2.5. Statistics

The data are expressed as mean ± standard deviation (SD). Because of skewed distributions, the L-carnitine-related metabolites and IMT were analyzed after logarithmic (log) transformation. The presence of CVD, DL, DM, HT, and habitual drinking or smoking was defined as 1, and their absence was defined as 0. Spearman’s ρ correlation coefficient and ANOVA were employed in univariate analyses between max-IMT or PS and other variables. Then, multiple linear regression (Model 1 for adjustment with the other metabolites and Model 2 for adjustment with all the variables) analyses were performed to examine whether there was an association between max-IMT or PS and metabolite levels. Further, the plasma levels of the carnitine-related metabolites in two different areas are presented in bar diagrams. The association between EPA and DHA with these metabolites is presented in a scatter plot. All statistical analyses were performed with the use of IBM SPSS Statistic software (SPSS Statistics 25). Statistical significance was defined as *p* < 0.05. 

## 3. Results

### 3.1. Demographic Data of the Studied Population

The baseline characteristics of the participants from the two different areas are shown in Table 1. The max-IMT and PS, as well as the incidence of CVD and HT, were all significantly greater in men than in women, while the incidence of DL was greater in women. The incidences of DL, DM, and HT were greater in the residents of Oki Island than in the residents of mountainous Kakeya county, in both men and in women. It is of note that both EPA and DHA in the serum were significantly greater in the residents of Oki than in the residents of Kakeya, which was probably due to the difference in fish consumption between the two regions.

Figure 1 summarizes the levels of L-carnitine-related metabolites in the residents of the two counties. When both areas were combined, the men showed significantly higher levels of L-carnitine, BB, and TML than the women, whereas there was no significant difference in plasma TMAO. Of interest, the plasma levels of TML and TMAO were significantly greater in the residents of Oki than in the residents of Kakeya, in both men and women, suggesting an influence of place of residence, probably due to the difference in daily diet, on TML and TMAO levels.

γBB and TML levels showed significant associations with age, while L-carnitine and TMAO levels were positively associated with body mass index (Appendix A). The levels of L-carnitine-related metabolites showed modest correlations with one another (Appendix A).

### 3.2. Simple and Multiple Regression Analyses of All Participants

In a univariate analysis in all participants (see Spearman’s ρ in Table 2), L-carnitine and γBB levels were correlated with max-IMT, whereas neither TMAO nor TML levels were correlated with max-IMT. A multiple linear regression analysis (Model 2 in Table 2) showed that age, sex, and hypertension were associated with max-IMT, whereas TML was inversely associated with max-IMT. No significant associations between either max-IMT and L-carnitine or max-IMT and TMAO were observed. When PS was substituted for max-IMT in the analysis, the inverse association of TML, as well as the positive association of age and sex (men over women), was replicated (see Model 2 in Appendix A). The association of L-carnitine or TMAO with PS was not significant (Appendix A).

### 3.3. Multiple Regression Analyses in Women and Men

To examine the potential sex difference in the effects of the L-carnitine-related metabolites, we tested the effects of the metabolites on max-IMT and PS separately in both the male and female subjects using a multiple regression analysis. In the female subjects, a significant association of L-carnitine with max-IMT, as well as with PS, was observed, whereas γBB, TMAO, and TML levels were not significantly associated (Table 3). This association of the L-carnitine level with max-IMT or with PS was not observed in men (Appendix A), suggesting that plasma L-carnitine level can only be a risk factor for atherosclerosis in women in this population.

### 3.4. Plasma Levels of TMAO and Other Metabolites Were Associated with the Plasma Concentrations of EPA and DHA

As fish is known to be rich in TMAO [18], the present observation of the greater TMAO level in the residents of Oki Island might be due to their greater consumption of fish. To examine this hypothesis, we performed an analysis to determine whether there was a correlation of plasma EPA/DHA with plasma TMAO/L-carnitine. As shown in Figure 2, although plasma L-carnitine correlated with neither plasma EPA nor plasma DHA (Figure 2A,B), plasma TMAO showed significant positive correlations with both plasma EPA and DHA (Figure 2C,D). There were no correlations observed between plasma EPA/DHA and γBB/TML levels (data not shown).

## 4. Discussion

In this cross-sectional study, we confirmed that TMAO is not associated with max-IMT or PS. As the TMAO level showed a positive correlation with EPA, as well as DHA, and as the TMAO level was higher in the residents of the island than in those in the mountainous area, fish was probably a major source of TMAO in the studied populations. In addition, we found for the first time that TML was inversely associated with IMT and PS, implying its protective role in atherogenesis in Japanese people. We also observed that the plasma level of L-carnitine was associated with max-IMT and PS, independent of age and hypertension, in women.

Although several studies have suggested that TMAO plays a causative role in atherogenesis [4,5,6,7], it has been reported that only γBB and L-carnitine levels are increased in patients with carotid atherosclerosis, whereas TMAO or TML levels remain unaffected [19]. Another clinical study failed to identify a positive association between TMAO and carotid atherosclerosis in young adults [27]. However, TMAO increased the risk of CVD in patients with particular backgrounds, such as those on dialysis [28] and those infected with HIV [29]. The role of TMAO in atherogenesis is therefore still controversial. Recently, plasma L-carnitine level became more of a focus since it was found to be associated with carotid atherosclerosis in a longitudinal study in patients infected with HIV [30]. The same study found that the progression of atherosclerotic plaque was associated not with basal levels of TMAO and choline but with betaine and L-carnitine. Moreover, after adjustment for confounding factors, adults who were infected with HIV and had high carnitine levels also had a significantly increased odds ratio (OR) for myocardial infarction, whereas no increased OR was found in the subjects with high levels of betaine, TMAO, or choline. Our results support these recent findings indicating no significant roles of TMAO and a potential role of L-carnitine in atherogenesis.

In this study, plasma TMAO levels in the residents of Oki Island were significantly higher than in those of the mountainous Kakeya area, and TMAO level was positively correlated with EPA and DHA. Recent studies have shown that plasma TMAO level is correlated with fish consumption [18,31,32,33]. In fact, fish consumption is high in both Japan and Norway, and the plasma TMAO level in our population is comparable to that in the population of Norway [20]. These findings suggest that a major source of TMAO in our population was fish. Indeed, our data from a questionnaire survey in 2019 regarding dietary intake frequency showed that Oki residents had a higher consumption of fish dishes than Kakeya residents (25% vs. 15% of more than once a day), as well as a lower consumption of meat dishes (13% vs. 4% of less than once a week). Although the consumption of vegetables was higher in Kakeya residents than in Oki residents (80% vs. 70% of more than once a day), the consumption of dairy foods and eggs was similar in the two areas. Although meat and eggs are, in general, important sources of TMAO, our result is further supported by the observation made by Krüger et al., who found that the association of plasma TMAO level with fish consumption was two times stronger than that with meat consumption [34]. 

Fish oil is a good source of long-chain ω-3 polyunsaturated fatty acids (PUFAs), mostly composed of EPA and DHA, which reduce the risk of CVD [35]. Many epidemiological studies have shown that fish consumption is associated with a lower risk of CV events [36,37,38] and that it reduces the mortality of heart failure [39], although this idea is not yet conclusive [40]. Inconsistency in the atherogenic role of TMAO among studies may therefore be due to differences in the source of TMAO, i.e., meat vs. fish; the deleterious effects of TMAO might be compensated for by PUFAs when fish is the major source of TMAO, while TMAO from meat might not have such a compensation. Another hypothesis is that TMAO may be a simple marker of meat (and fish) consumption and that L-carnitine or other metabolites from meat is the true culprit. Further studies are necessary to clarify the influence of food composition on the pathophysiological roles of L-carnitine-related metabolites. 

In our analyses, we also observed that TML showed an inverse association with IMT and PS, which is inconsistent with previous reports [21,22,23]. TML is known to be a major precursor of L-carnitine in the de novo synthetic pathway, and, therefore, it is thought to have a deleterious effect on atherosclerosis similarly to L-carnitine [41,42,43]. According to the study conducted by Servillo et al., however, one of the major sources of TML is vegetables, which are essential components of a healthy diet [44]. As vegetable proteins, such as casein, soy protein, and wheat gluten, have been discovered to be sources of TML [45], TML may have acted as a marker of vegetable consumption in our population. Vegetables have been assumed to prevent the progression of atherosclerosis [45,46] via the reduction of plasma lipids, antioxidant effects, antiproliferative and anti-migratory effects on smooth muscle cells, and the maintenance of normal vascular reactivity [47,48]. As in the case of TMAO and fish consumption, we may need to be cautious about the dietary sources of TML when considering its pathophysiological roles. Nevertheless, the inverse association of TML with atherosclerosis needs to be confirmed in other populations.

We observed a sex difference in the association between plasma L-carnitine and carotid atherosclerosis. This sex difference might be based on differences in the composition of gut microbiota, in the immune function, in the vascular function, or in other unknown mechanisms [49,50]. 

The precise mechanisms underlying how L-carnitine and/or its metabolites deteriorate atherosclerosis have not yet been clarified. L-carnitine plays a pivotal role in the mitochondrial membrane transport of fatty acid and, thus, may affect mitochondrial function [51]. Although potential pro-inflammatory roles of L-carnitine have been suggested, Sinha et al. showed no correlations of L-carnitine levels with interleukin-6, d-dimer, and high-sensitive CRP [30]. On the contrary, plasma TMAO level has been positively associated with low-grade inflammation in a German population [52]. Detailed biochemical and/or cell biological studies are necessary to address this issue.

There are a couple of limitations to this study. First, because this is a cross-sectional study, causal relationships cannot be inferred. Second, we could not control for the effects of unmeasured factors, such as fish consumption, active lifestyle, sedentary behavior, and occupational status. Accordingly, the correlation between L-carnitine-related metabolites and the consumption of various food could not be analyzed. Third, the small-sample bias is present. Although we showed no significant association between TMAO and IMT, future studies with larger sample sizes are necessary to conclude this issue. Finally, we did not evaluate the inflammation levels in the participants. Hence, we could not examine the relationship between inflammation status and carotid IMT, as well as the metabolites. Despite these limitations, this study provides new evidence about the association between L-carnitine-related metabolites and atherosclerosis using a unique set of participants, i.e., those from an isolated island and those from a mountainous area, implying the importance of the dietary source of L-carnitine-related metabolites when the effects of those metabolites are considered.

## 5. Conclusions

In this cross-sectional study, we found that plasma TMAO level was not a risk factor for atherosclerosis in an elderly population of Japan and that fish was a major dietary source of TMAO. We assume that consuming fish will reduce the impacts of TMAO on atherosclerosis and cardiovascular disease. In addition, plasma TML level was inversely associated with atherosclerosis, whereas plasma L-carnitine level was a significant marker for atherosclerosis in older women. Future studies are necessary to confirm our results in different populations and to establish the causal relationship between these metabolites and atherosclerosis using a longitudinal study design. 

## Figures and Tables

**Figure 1 nutrients-15-00759-f001:**
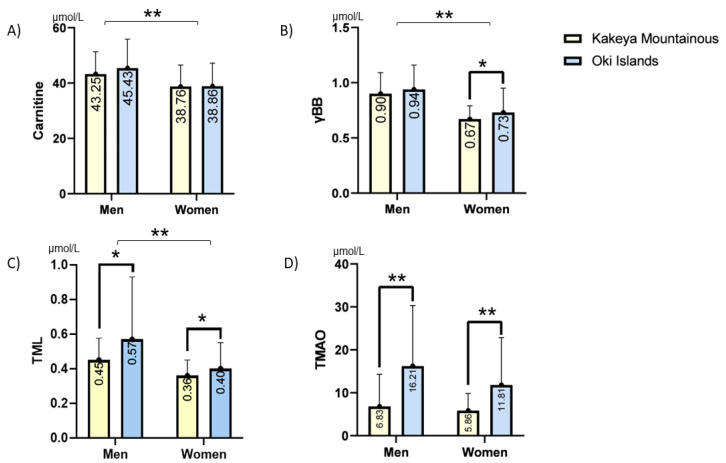
Plasma L-carnitine (**A**), γBB (**B**), TML (**C**), and TMAO (**D**) levels in the two populations. The four metabolites were measured using liquid chromatography–tandem mass spectrometry as described in the Materials and Methods Section. The white and blue columns show the levels in the populations of Kakeya and Oki, respectively. Each column and error bar indicates mean and SD, respectively. The levels of L-carnitine, γBB, and TML in men were significantly greater than those in women. * *p* < 0.05 and ** *p* < 0.001 using Student’s *t*-test.

**Figure 2 nutrients-15-00759-f002:**
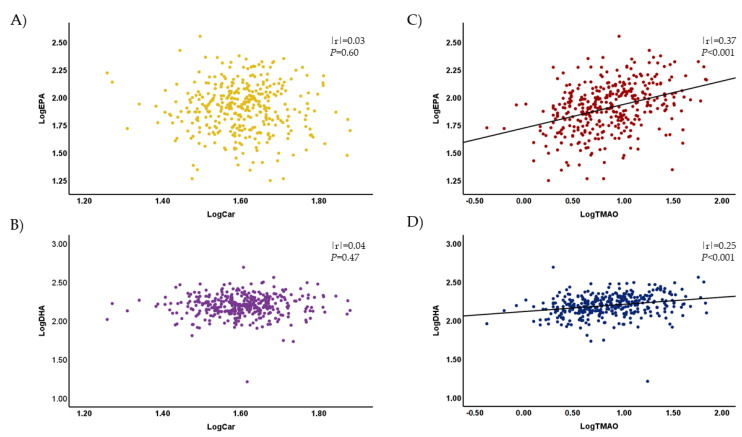
Correlation of plasma L-carnitine and TMAO with plasma EPA and DHA. Plasma L-carnitine did not correlate with plasma EPA (**A**) or DHA (**B**), but plasma TMAO significantly correlated with both plasma EPA (**C**) and DHA (**D**). TMAO and L-carnitine were measured using liquid chromatography–tandem mass spectrometry, and EPA and DHA were measured using gas chromatography–mass spectrometry as described in the Materials and Methods Section. Log-transformed values were analyzed using Pearson’s correlation coefficient.

**Table 1 nutrients-15-00759-t001:** Background characteristics of the 364 participants.

	Total (n: 364)			Men (n:142)			Women (n:222)		
Variables	Men (n: 142)	Women (n: 222)	*p*-Value	Kakeya (n: 72)	Oki (n: 70)	*p*-Value	Kakeya (n: 85)	Oki (n: 137)	*p*-Value
Age (y)	73.0 ± 8.5	72.5 ± 7.2	0.512	73.1 ± 8.4	73.0 ± 8.7	0.961	72.7 ± 6.3	72.3 ± 7.8	0.665
Height (cm)	162.6 ± 6.7	149.6 ± 5.7	<0.001	163.8 ± 6.5	161.5 ± 6.8	<0.05	150.1 ± 5.7	149.2 ± 5.8	0.312
Weight (kg)	61.5 ± 10.3	50.8 ± 8.7	<0.001	61.1 ± 10.1	62.0 ± 10.5	0.566	48.2 ± 8.0	52.4 ± 8.7	<0.001
BMI (kg/m^2^)	23.2 ± 3.2	22.7 ± 3.5	0.158	22.7 ± 3.2	23.7 ± 3.1	0.064	21.4 ± 3.3	23.5 ± 3.5	<0.001
DL (%)	15.5	35.1	<0.001	15.3	15.7	0.218	28.2	39.4	<0.05
DM (%)	10.6	7.2	0.264	8.3	12.9	<0.05	7.1	7.3	0.653
HT (%)	52.8	40.1	<0.05	47.2	58.6	<0.05	36.5	42.3	0.120
CVD (%)	9.64	5.23	<0.001	9.55	9.71	0.288	3.18	6.8	0.264
Habitual drinking (%)	44.4	6.3	<0.001	47.2	41.4	0.491	9.4	4.4	0.135
Current smoking (%)	8.5	1.4	<0.001	9.7	7.1	0.584	1.2	1.5	0.860
EPA (μg/mL)	95.9 ± 56.7	87.82 ± 38.6	0.108	81.2 ± 45.2	111.1 ± 63.31	0.002	73.3 ± 32,0	96.9 ± 39.6	<0.001
DHA (μg/mL)	164.21 ± 63.4	166.42 ± 43.82	0.695	142.9 ± 41.1	186.2 ± 74.2	<0.001	143.8 ± 32.8	180.4 ± 44.0	<0.001
Max-IMT (mm)	2.08 (1.92–2.24)	1.74 (1.64–1.84)	<0.001	1.92 (1.72–2.11)	1.98 (1.74–2.22)	0.706	1.75 (1.59–1.92)	1.73 (1.61–1.85)	0.861
PS (mm)	5.42 (4.63–6.21)	3.26 (2.90–3.61)	<0.001	4.81 (3.79–5.83)	4.81 (3.67–5.96)	0.996	3.18 (2.67–3.70)	3.31 (2.83–3.78)	0.483

Data are expressed as the mean ± SD or the mean (95% confident interval). Chi-square test was applied for the incidences of Dl, DM, HT, CVD, habitual drinking, and current smoking. Student’s *t*-test was applied for other parameters. DL; dyslipidemia, DM; diabetes mellitus, HT; hypertension, CVD; cardiovascular disease, IMT; intima–media thickness, PS; plaque score.

**Table 2 nutrients-15-00759-t002:** Simple and multiple linear regression analyses of IMT in all participants.

Variables	Spearman’s ρ		Model 1						Model 2					
	ρ	*p*-Value	β	SE	Standard β	t	*p*-Value	VIF	β	SE	Standard β	t	*p*-Value	VIF
Carnitine	0.122	0.020	0.009	0.005	0.096	1.725	0.085	1.152	0.006	0.005	0.072	1.312	0.190	1.274
γBB	0.162	0.002	0.337	0.221	0.091	1.527	0.128	1.327	−0.120	0.232	−0.033	−0.516	0.606	1.664
TMAO	−0.018	0.726	0.004	0.005	0.048	0.793	0.428	1.351	0.008	0.005	0.107	1.759	0.080	1.543
TML	0.061	0.249	−0.069	0.228	−0.018	−0.303	0.762	1.280	−0.588	0.228	−0.152	−2.581	0.010	1.447
Age	0.314	<0.001							0.029	0.006	0.279	5.138	<0.001	1.234
BMI	0.008	0.880							−0.007	0.013	−0.030	−0.550	0.582	1.223
DL	−0.044	0.402							−0.003	0.088	−0.002	−0.037	0.971	1.174
DM	0.033	0.536							0.030	0.123	0.013	0.246	0.806	1.096
HT	0.257	<0.001							0.202	0.089	0.122	2.264	0.024	1.212
Drinker	0.107	0.041							0.041	0.111	0.021	0.369	0.712	1.330
Smoker	0.014	0.791							0.088	0.205	0.022	0.427	0.670	1.074
Sex #1	0.214	<0.001							0.364	0.109	0.221	3.328	0.001	1.839
Area #2	−0.014	0.785							0.001	0.088	0.001	0.012	0.991	1.236

In Model 1, only the L-carnitine-related metabolites were included in the analyses. In Model 2, all the parameters shown in the left column were included. SE; standard error, VIF; variance inflation factor. #1. 0 = women, and 1 = men, #2. 0 = Kakeya, and 1 = Oki Island.3.2 Simple and multiple regression analyses in all participants.

**Table 3 nutrients-15-00759-t003:** Multiple linear regression analysis of max-IMT and PS in women.

Variables	IMT						PS					
	β	SE	Standard β	t	*p*-Value	VIF	β	SE	Standard β	t	*p*-Value	VIF
Carnitine	0.017	0.006	0.204	2.941	0.004	1.207	0.066	0.022	0.213	3.015	0.003	1.207
γBB	−0.395	0.281	−0.108	−1.405	0.161	1.472	−0.851	1.049	−0.063	−0.811	0.418	1.472
TMAO	0.009	0.007	0.125	1.383	0.168	2.047	0.023	0.024	0.087	0.942	0.347	2.047
TML	−0.826	0.454	−0.153	−1.819	0.070	1.789	−3.172	1.694	−0.161	−1.873	0.063	1.789
Age	0.028	0.007	0.294	4.199	<0.001	1.230	0.100	0.025	0.289	4.051	<0.001	1.230
BMI	−0.006	0.014	−0.031	−0.434	0.665	1.296	0.005	0.052	0.007	0.089	0.929	1.296
DL	−0.093	0.092	−0.068	−1.013	0.312	1.116	−0.221	0.341	−0.044	−0.646	0.519	1.116
DM	−0.080	0.150	−0.034	−0.529	0.598	1.044	0.182	0.561	0.021	0.324	0.747	1.044
HT	0.239	0.097	0.172	2.478	0.014	1.206	0.428	0.360	0.084	1.189	0.236	1.206
Drinker	−0.012	0.185	−0.004	−0.065	0.948	1.091	0.130	0.691	0.013	0.188	0.851	1.091
Smoker	0.050	0.385	0.008	0.129	0.898	1.063	0.966	1.435	0.045	0.673	0.502	1.063
Area #1	0.010	0.099	0.007	0.099	0.921	1.243	0.264	0.369	0.051	0.716	0.475	1.243

SE; standard error, VIF; variance inflation factor. #1. 0 = Kakeya, and 1 = Oki Island.

## Data Availability

Not applicable.

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
