# Peer review of "Neither Trimethylamine-N-Oxide nor Trimethyllysine Is Associated with Atherosclerosis: A Cross-Sectional Study in Older Japanese Adults"

_nutrients, 2023, doi:10.3390/nu15030759_

Round 1

Reviewer 1 Report

First of all, thank you for inviting me to review the paper “Neither trimethylamine-N-oxide nor trimethyllysine is associated with atherosclerosis: a cross-sectional study in aged Japanese”. Study is quite interesting, wherein the notion of having fish as a major dietary source of TMAO in Japan is somehow not an apparent risk for atherosclerosis.

Some recommendations and questions are provided:

Abstract is adequate, should adhere to journal guidelines for length of abstract.

Introduction

Line 63 to 65 – recheck or maybe rephrase for “red meat….. “ line 65, might “deteriorate” CVD through the …., just a little confused, was thinking the overconsumption of red meat would produce TMAO, thus increase the chance of CVD.

Might be helpful to provide a hypothesis (at the end of the introduction)

For the subjects, might also be helpful if some statistics on the Kakeya and Oki residents, their livelihood, or some data on fish consumption, and might also be helpful to provide some contrast with the general Japanese population. In addition, the occupation of the residents might also be an issue, such as a more active lifestyle as against city folks who spend time in seating in offices.

The statistical analysis is adequate. Just a question, were controlled variables used? Why and why not?

Could help if the authors could provide some practical implications. Sort of what now?

Author Response

Thank you for your valuable propositions. Your thoughtful comments and recommendations on our manuscript will improve it a lot.

Q1] Abstract is adequate, should adhere to journal guidelines for length of abstract.

Response: Thank you for pointing out these very important aspects. We have tried to follow the journal guidelines for the length of the abstract. Please see page 1, lines 24-47.

Q2] Line 63 to 65 – recheck or maybe rephrase for “red meat….. “ line 65, might “deteriorate” CVD through the …., just a little confused, was thinking the overconsumption of red meat would produce TMAO, thus increase the chance of CVD.

Response: According to your suggestion we checked this line into our manuscript and rephrase it. Please see page 2, line 67.

Q3] Might be helpful to provide a hypothesis (at the end of the introduction).

Response: Thank you very much for your kind suggestion. We have added a hypothesis at the end of the introduction. Please see page 2, lines 83-84.

Q4] For the subjects, might also be helpful if some statistics on the Kakeya and Oki residents, their livelihood, or some data on fish consumption, and might also be helpful to provide some contrast with the general Japanese population. In addition, the occupation of the residents might also be an issue, such as a more active lifestyle as against city folks who spend time in seating in offices.

Response: Thank you for the thoughtful question. We added our questionnaire results in the discussion in page 9, lines 259-265. 

”Indeed, our data from a questionnaire survey in 2019 regarding dietary intake frequency showed higher chance of fish dishes in Oki residents, compared to Kakeya (25% vs 15% of more than once a day), and lower chance of meat dishes (13% vs 4% of less than once a week). Although higher chance of taking vegetables in Kakeya residents compared to Oki residents (80% vs 70% of more than once a day), chance of taking dairy foods and eggs was similar in two areas.”

In addition, according to recent data from the government office, people participating agriculture/forestry and fishing are 3.3% and 21.2% in Oki (#1), whereas 35% and 0%, respectively in Kakeya (#2). These findings also support our hypothesis, where fish is a major source of TMAO in Japanese population.

#1. http://www.machimura.maff.go.jp/machi/contents/32/528/details.html

#2. https://www.city.unnan.shimane.jp/unnan/shiseijouhou/jouhoukoukai/toukei/toukei.html

[Q5] The statistical analysis is adequate. Just a question, were controlled variables used? Why and why not?

Response: Thank you for these important questions. We did not use controlled variables in this study. We added a sentence in the limitation (page 9, lines 308-310).

“We could not control for the effects of unmeasured factors, such as fish consumption, active lifestyle, sedentary behavior, and occupational status.”

[Q6] Could help if the authors could provide some practical implications. Sort of what now?

Response: According to your suggestions, we included practical implications in our manuscript. From our research, “We assume that taking fish will reduce the impact of TMAO on atherosclerosis and cardiovascular disease.” Please see page 10, lines 325-326.

Reviewer 2 Report

This paper is interesting and well written. The only observation will have, and we consider that this should be reflected in the limitations section is the small sample bias. Whereas the Spearman correlation showed no significant correlation between TMAO and IMT, Model 2 shows a p value of 0.08 for TMAO. Now, p values are not magical binary tools, but rather are part of a continuum, so that 0.08 is close to the conventional threshold of 0.05, and with a higher sample size, the multivariate analysis could conclude that TMAO is significantly associated with IMT. In our view, this aspect should be acknowledged in the limitations section, because the sample size was relatively small.

Author Response

Thank you very much for your valuable comments. We appreciate your thoughtful suggestions.

In the limitations section, we acknowledge that with larger sample size is necessary to conclude this issue. Please see page10, lines 312-315.